# The Problem of Reliability in Public Transport for the Metropolis GMZ Area-Pilots Studies

**Agnieszka Gaschi-Uciecha** 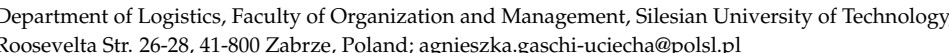

Department of Logistics, Faculty of Organization and Management, Silesian University of Technology, Roosevelta Str. 26-28, 41-800 Zabrze, Poland; agnieszka.gaschi-uciecha@polsl.pl

**Abstract:** Ensuring the sustainable development of transportation whether it involves a country, a city or a metropolitan area is becoming a priority for modern Europe. Therefore, the development of sustainable transportation is crucial in cities. It should aim to increase the number of trips made by public transportation while minimizing social costs and negative environmental impacts. The intensive development of cities from industrial to service-based and the phenomenon of suburbanization require changes in public transport services. Often, however, carriers do not change existing lines and stops and schedules justifying this by passenger habits. This approach can cause a mismatch between the availability of public transportation services and the demand for these services. Therefore, planning and improving the public transportation system should be based on careful analysis. There are different approaches to this problem. The article uses the Customer Satisfaction Index (CSI) method to assess service quality, including reliability, where the main measurement tool was a survey questionnaire. A pilot studies has been conducted to present the opinion of residents of the Górnośląsko-Zagłębiowskiej Metropolis (GZM) on specific parameters of public transportation. The GZM is a large metropolitan area in the Silesian province, consisting of 41 municipalities. The metropolitan area is inhabited by more than 2.3 million people and covers an area of about 2550 square kilometers. The results of the pilot studies are a contribution to further research and may be useful to understand the essence of reliability in public transport.

**Keywords:** public transport; reliability; Customer Satisfaction Index (CSI)

## 1. Introduction

Despite a wealth of literature, research and experience, the issue of urban public transportation is still not fully explained.

Public transportation plays a very important role in the functioning of modern cities. Not only does it help meet the transportation needs of residents, but it is also a key component of a city's transportation system [1].

Studies show that satisfaction with the living conditions of residents of an area is related to the quality of travel within the area [2]. However, mobility in large urban metropolitan areas faces a number of problems, i.e., congestion manifested by lack of parking spaces or congestion during rush hour, noise, as well as air pollution and the problem of parking availability [3,4]. Most of these problems are due to the increasing number of individual transport users. The way to solve them is to change the way city residents think about public transportation and encourage them to use it more often [5], which in turn will translate into a gradual increase, not only in importance, but also in popularity of public transportation—including among motorized residents.

In order to convince users to switch to public transportation, the needs and expectations of residents must be met [6]. This is especially important in the situation after the COVID-19 epidemic, when the share of trips by public transport has decreased significantly [2,7]. Improving the use of public transport is supported by the provision of a modern fleet with vehicles equipped with environmentally friendly engines and meeting

passengers' quality expectations, such as air conditioning, smartphone chargers, free Wi-Fi, and accurate route information [8].

The dynamic growth of urban areas is forcing public transport operators to make changes to ensure accessibility to public transportation [9,10]. These measures are necessary to ensure greater access to public transportation, which will ensure demand for transportation services and reduce traffic exclusion [11,12]. These solutions are in line with the idea of sustainable development and environmental trends [13]. The availability of public transportation services increases mobility, and can be a factor in accelerating urban development [6].

An efficient and reliable transportation system is necessary to provide conditions for the development and territorial expansion of the city. In this aspect, however, it is necessary to adapt to the needs and preferences of customers.

The peculiarities of urban transportation include both tangible and intangible elements of the service provided, which can be perceived and evaluated differently by passengers. Customer satisfaction is of great importance here. Therefore, companies providing transportation services should, first of all, continuously monitor the needs and expectations of customers and verify their satisfaction.

Various approaches to this problem can be found in the literature, including survey methods [6].

This article presents the results of a pilot study conducted using the Customer Satisfaction Index (CSI) method to learn about and assess passenger opinions on the reliability of public transportation operating in the Górnosląsko-Zagłębiowskiej Metropolis area, including the degree to which transportation demands are met. In addition, the pilot studies provides a basis for further research in this context.

The article is divided into five parts. Section 2 presents a literature review, which presents the role of transportation in the functioning of the city. In addition, attention is paid to the transportation needs of local residents and the importance of reliability in public transportation is introduced. In Section 3, the research method of reliability assessment is presented along with the research sample. The next chapter presents the results of the pilot study. Conclusions are presented in Section 5, respectively.

## 2. Theoretical Background

**The role of transport in city functioning**. Transport has been and continues to be an important factor of change in today's world, which expects, on the one hand, opportunities for the fast, safe and economic movement of people and goods and, on the other hand, a high quality of service in this respect [14]. It can be argued that it accompanies every economic and social activity. Moreover, it is an indispensable condition for determining economic development. Transport cannot be replaced by other activities or processes; there is no substitute for transport activities [15].

The city's transport system is a catalyst for both economic and social opportunities that stimulate urban efficiency and productivity. Moreover, through its complementary nature and links with the social and economic environment, it is a factor integrating and coordinating the urban economy becoming a universal and irreplaceable element of economic processes as well as manifestations of social life occurring within urban agglomerations [14].

The role of transport in a city is not only limited to supporting production or catalysing economic growth, but directly affects the size of the area a city can cover [14]. This is because the spatial area of a city is closely dependent on the transport accessibility provided by the transport modes designed to serve the city. However, there is always a limit at which travel time restricts the distance that passengers are willing to travel by a given mode of transport. In practice, therefore, the spatial development of a city occurs until the means of transport serving it reaches the range limit, understood as the distance that passengers can cover with a particular means of transport in the time they are able to devote to travel [16].

A similar view is expressed by B. Rzeczyński [17], who argues that the transport system in cities is developed not so much to shorten the travel time between its individual components, but precisely to enable further territorial expansion of the city. The author [4] concludes from historical analyses of urban development that the development of transport and city space are correlated in such a way as to ensure that residents can cover a distance equal to the radius of the city in approximately 30 min.

The above claims are consistent with the principle of a 'fixed budget of time wasted on transport' formulated by Y. Zahavi [18], according to which there is a tendency among passengers nowadays to devote a fixed part of the day to travel, irrespective of their ability to reach their destinations. This means that city dwellers are able to spend a fixed amount of time travelling, e.g., between work and home, regardless of the possibilities offered by the means of transport, which means that as a city's transport system develops, its effective range changes, as passengers can cover longer distances in the same amount of time they are willing to spend moving around the city. It should be borne in mind, however, that travel time consists not only of the journey, but also of getting to the means of transport and (in the case of public transport) waiting for the vehicle to arrive at the stop, possible transfers and getting from the vehicle to the destination [19].

Transport plays a very important role in the functioning of a city. The level of development of the transport system operating in the city and its environs is one of the criteria determining the possibilities of economic growth of the region and conditioning the possibility and scope of the possible territorial expansion of the city. By integrating the other functional areas of the city, transport ensures their efficient operation and enables the needs of the inhabitants to be met. Due to its complementary and integrative nature, transport is of key importance in ensuring the proper development of the city and the quality of life of its inhabitants.

**Transport needs of city residents**. The importance of the communication need is very important when considering transport in the city. O. Wyszomirski [20] defines a communication need as "the desire, need or demand of an individual or a specific collectivity to carry out the process of moving from one place to another". Similarly, according to M. Szymczak [21], who treats a communication need as the need to make a journey from a starting point to a destination at a specific point in time. In turn, G. Zimon and B. Gosik [22] claim that a communication need is a desire or necessity to move from one place to another by means of transport expressed by an individual or a specific collective. Based on the examples cited, it can be concluded that communication need is equated in the literature with the need for people to move from place A to B, but the concept of communication itself does not refer only to transportation.

Communication needs are secondary, meaning that they are triggered by the need or desire to satisfy other needs. Three dimensions of communication needs can be distinguished [17]:

Quantitative dimension—refers to the number and length of trips.

Spatial dimension—is expressed in terms of a displacement vector.

Temporal dimension—is determined by the date and time of travel, as well as the timetable of journeys.

Transport needs only represent potential demand, as the mere occurrence of a need does not yet mean that a person will actually buy a ticket and realise his or her intention to travel to the desired destination. For this to happen, the right conditions must be created in the form of an attractive transport offer by urban public transport. The transport offer includes not only the ticket price (or fare), but also, among other things, the timetable, the fleet owned by the operator, the ease of access to information regarding the service and the safety offered by the operator. For potential passengers to be willing to consider a transport offer, it should take all their preferences into account. However, it is important to bear in mind that it is not possible to attract all social groups at the same time, as some people will not be interested in using urban public transport services regardless of the quality of the transport offer, e.g., because they prefer individual transport [23].

**The importance of reliability in public transport.** The concept of reliability is used in various industries and is thus considered from many perspectives. According to T. Nowakowski [24], reliability consists of properties such as readiness and credibility (Figure 1).

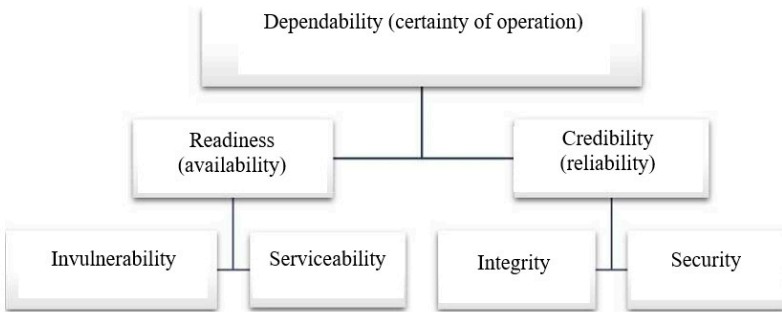

**Figure 1.** Diagram of the reliability concept hierarchy. Source: [18].

J. C. Laprie [25] defines reliability as the ability of a system to deliver a specific service. And J. Twaróg [26], on the other hand, interprets the reliability of the logistics system on this basis as ensuring timely and uninterrupted delivery of specific products. The complexity of the problem of defining reliability is highlighted by many authors. A. Avizienis J.C. Laprie and B. Randell [27] state that in order to properly define the concept of reliability, it should be considered from three points of view. Namely, it is necessary to consider the threats to reliability, its characteristics (attributes) and ways to achieve it.

Thus, A. Tubis and S. Werbińska-Wojciechowska [28] consider the reliability of the passenger transport system as "the power to ensure the possibility of travel from place A to place B in a certain time and conditions of use". Thus, reliability in the sense of the transport system does not refer only to the failure rate of the means of transport (with which it is sometimes equated), but is a much broader concept, since in addition to the requirement for the movement from point A to point B itself, it also requires punctuality and the provision of appropriate travel conditions. On this basis, it can be said that reliability in public transport means the provision of transport services in accordance with established contractual conditions, such as timetables, and with the fulfillment of user requirements, which in the literature are referred to as transport demands. The measure of such assumed reliability of public transport can be, for example, the percentage of correctly (according to the timetable) performed transport operations (trips) in relation to all performed or contracted operations [29].

Analyzing the literature on collective public transport, it can be noted that, in addition to the mere realization of the purpose of the trip, which is to move people from place A to place B, great attention is paid to the study of travel time. This is definitely a very important element of reliability that follows directly from its definition. In the literature, this element is referred to as travel time reliability and talks about the deviation between planned and actual travel time and its repeatability for trips that are repetitive (such as commuting) [18]. From the passenger's point of view, this is indeed very important, because in the case of a reliably operating transportation system, the passenger can predict in advance how long a trip will last. Another area of time unreliability is delays of transport means, i.e., deviations of the actual time of arrival at the stop of a transport means from the time assumed in the timetable [9,30].

In addition, passengers lose a lot of time getting to their destinations via public transportation due to imperfect service by some transportation companies [6,31].

Many transportation companies focus their attention on scheduling departure and arrival times, taking into account some basic obstacles such as public road congestion, accidents and others, which are often not enough to reach the required destination [6].

The recurrence of this problem has attracted the interest of researchers to find alternative solutions and more comprehensive terminology to determine the factors affecting public transportation service [6,32].

On the other hand, a number of studies have divided the reliability of the service into several levels, ranging from the city level to the block and stop level [7,10].

These factors have been classified, among others. into internal factors e.g., control of the number of passengers, quality of service, pricing and other factors that are often under the control of transport companies, and external factors that are under the control of others and cannot be controlled by transport companies [33]. Changes and improvements in these factors directly affect passenger satisfaction and loyalty, which in turn translates into transportation reliability.

S. Iftekhar and S. Tapsuwan, [34] provided an overview of the of factors influencing traveler behavior and transportation choice. M. Janic [9] reviewed reliability research programs that have been implemented for public transportation in the European Union.

Summarizing the literature review, it can be concluded that issues related to transportation reliability have not been studied in depth. Moreover, as stated by M. A. Alkubati et.al. [6], according to the WOS database, nine review papers have been published on public transportation reliability.

This article presents the results of a pilot study on finding out the opinions of public transportation passengers on its reliability. In addition, the results serve as a first step for further in-depth studies in this field.

## 3. Methodology

There are a number of methods and tools that can be used to study public transportation [4,35], including for reliability. Most of these studies are based on the use of the Servqual method [12,36,37], indicator methods [38] or other extended or modified methods [39].

In addition, research by T. Chuenyindee et al. [40] focused on SERVQUAL dimensions and substantiated the need to use other methods to assess service quality [40], which in turn translates into reliability in transportation. Reliability, in turn, has a significant impact on public transportation users' satisfaction with the services they receive.

For the purposes of this study, particular attention was paid to the CSI (customer satisfaction index) method.

The CSI method, the Customer Satisfaction Index, allows measuring the level of customer satisfaction in terms of selected attributes, which in turn translates into the reliability of transportation, including public transportation. The Customer Satisfaction Index is calculated on the basis of a weighted score, which consists of the rating of individual elements and the weight assigned to them [41].

The application of the CSI method requires the preparation of a survey questionnaire, in which respondents, in addition to evaluating individual service elements, also assign them a weight according to their own feelings [41].

The purpose of the survey questionnaire was to obtain data on the evaluation and importance that passengers assign to each of the surveyed postulates. For this purpose, a questionnaire was prepared with 47 questions, including 46 choice questions divided into 3 sections.

The first section consisted of a metric and questions designed to determine the frequency and nature of the public transportation trips made by the respondent.

The second section was designed to find out the importance of each transportation postulate (19) to passengers. Respondents were asked to rate the importance of each postulate on a five-point Likert [35] scale, where: 1 represented the least importance and 5 the highest.

The third section assessed the current, perceived state of implementation of the postulates whose importance (significance) they determined in Section 2. The evaluation was also made according to a 5-point scale.

The last question was open-ended and not required to be completed. Repeat respondents were allowed to post comments on the survey, as well as elaborate on aspects related to the research topic that were not included in the questionnaire. These responses were not

included in the calculation of the passenger satisfaction index, but their comments may be helpful at a later stage of the research.

The survey was conducted online by providing a link to an interactive form created for the survey using free Google Forms software. The link to the survey was posted on a popular social networking site, groups related to public transportation operating in the GZM, groups for students and graduates of the Silesian University of Technology, news groups of selected GZM cities, and the official passenger group—the Public Transport Authority (PTA). The collection of respondents' opinions lasted for six weeks, running from 29 April to 7 June 2021. During this period, 108 declarations were received, of which 101 were completed correctly. The survey was a pilot and is the premise for a baseline study. P. B. Sztabinski [42] says that pilot surveys are conducted on a small sample of 20–50 people, while K. Grzeszkiewicz-Radulska [43] indicates that pilot surveys involve 30–100 respondents. Therefore, 101 correctly completed forms were considered for analysis.

A noticeable majority of respondents (almost 58%) were men. This may be due to the fact that they make up a larger proportion of the members of public transport enthusiast groups.

More than half of the survey members were in the age group between 18 and 26. In contrast, one in five people declared an age between 26 and 40, making them the second most numerous age group in the survey. The next largest age group in terms of participation was under 18 years of age, with almost 14%, followed by those aged between 40 and 60 (just under 10%). The least numerous group, on the other hand, were older people aged 60 and over.

Of the participants in the survey, almost 40% of people do not own a personal car, slightly less, at just under 31%, were those who also do not own a car but have the option of using it at will (e.g., company car, parents' car, husband's car, wife's car, children's car, etc.).

## 4. Results and Discussion

In order to get to know the opinions of passengers on the reliability of public transport operating in the Górnośląsko-Zagłębiowskiej Metropolis including the degree of implementation of transport demands, the CSI research method, i.e., the Customer Satisfaction Index, was used.

**Section 1.** A noticeable majority of respondents (almost 58%) were men. This may be due to the fact that they constitute the greater part of the members of groups associating public transport enthusiasts. The other groups to which the survey was addressed are characterized by a relatively equal division by gender.

More than half of the survey members were people between the ages of 18 and 26. Every fifth person declared the age between 26 and 40, which makes them the second largest age group in the study. The next largest age group in terms of shares were people under 18 with a result of almost 14%, followed by people aged 40 to 60 (less than 10%). The least numerous group were elderly people over 60 years of age.

However, a large variation was registered in the case of education. The dominant groups of respondents were people with secondary and higher education with a title, constituting less than 39% and 36% of the survey members, respectively. Primary education was declared by about 13% of the participants, while higher education, but without a title, was declared by only 9% of the respondents. The smallest group were people with vocational education, constituting 4% of the respondents.

Among the survey participants, almost 40% of people do not have their own car, slightly less, because less than 31% were people who also did not have their own car, but could freely use it (e.g., a company car, parents, husband, wife, children, etc.). The remaining, smallest part of the respondents were people who had their own car.

The next questions concerned the frequency and nature of public transport trips made by the study participants. The largest groups of respondents were people using public transport every day (26%) and several times a week (22%). A slightly less numerous group were people traveling by public transport several times a month (19%) and several times a year (13%). The least numerous groups were people traveling more than twice a day (10%)

and once a month (7%). Complete lack of using public transport services was declared by only 4% of respondents.

It can be seen that more than half of the respondents (58% to be exact) use public transport at least several times a week, so they have daily or almost daily contact with its services. If we add people traveling at least several times a month to this, it turns out that as many as 77% of the respondents had at least occasional contact with public transport, which ensures relatively high reliability of the research results. The detailed distribution of individual groups of respondents is shown in Figure 2.

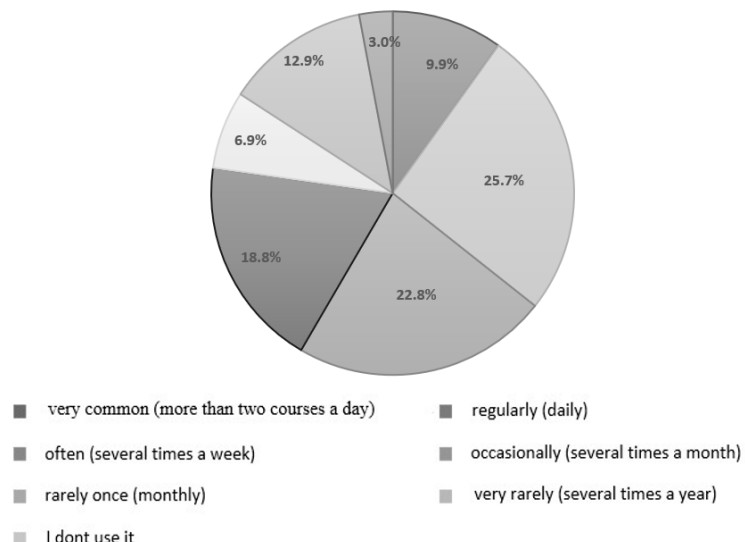

**Figure 2.** Percentage of respondents' answers. Source: Elaboration based on K. Rusinek diploma thesis [2021].

The last question concerned the frequency of a specific type (in terms of duration) of travel by the respondents. Participants of the study had to specify how often they use public transport for short (up to 20 min), medium (20–40 min) and long (more than 40 min) journeys. Respondents were to include in the travel time not only the time of travel by means of transport, but also all transfers included in a given transport journey and waiting for means of transport. The answers of the respondents are shown in Figure 3.

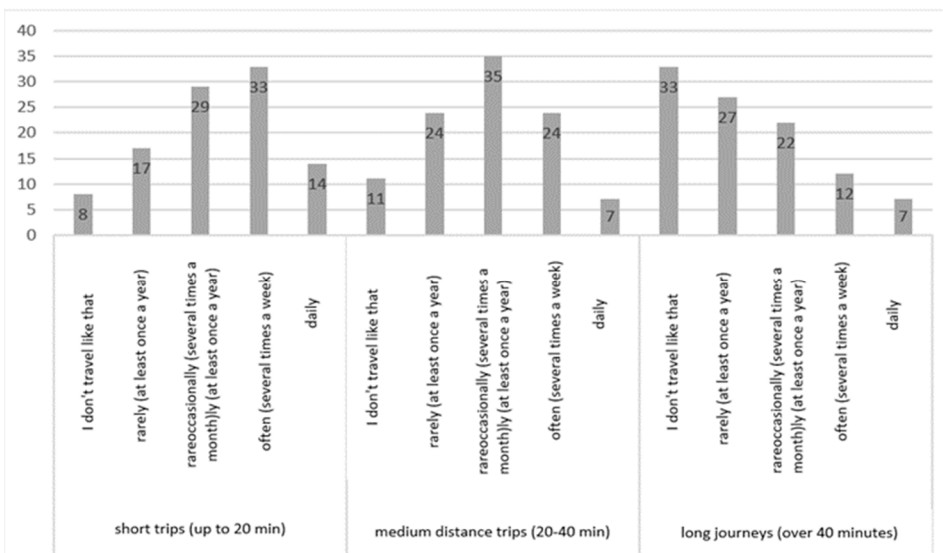

**Figure 3.** Respondents' answers to the question regarding the frequency of individual types of travel divided by time. Source: Elaboration based on K. Rusinek diploma thesis [2021].

From the analysis of the above graph, it can be concluded that most trips made by public transport are short trips lasting up to 20 min. Journeys longer than 20 min are less frequent. Trips longer than 40 min are rare, as many as 33 out of 101 respondents do not do it at all, and another 27 do it rarely. It can be concluded that the longer the journey is supposed to last, the less frequently the inhabitants choose public transport.

**Section 2.** In the second part of the questionnaire, the respondents assessed the importance of individual parameters of public transport, giving them weights from 1 to 5, where 1 means no importance and 5 means very important. The answers of the respondents are summarized in Table 1

**Table 1.** Summary of the importance of the tested parameters for passengers and the calculated average weight.

| Public Transport Parameter/Feature | Respondents' Answers (Parameter Weights) | | | | | |
|---|---|---|---|---|---|---|
| | **1** | **2** | **3** | **4** | **5** | **Importance for Passengers** |
| Certainty of reaching the destination | 2 | 0 | 5 | 18 | 76 | 4.64 |
| Punctuality | 2 | 1 | 4 | 27 | 67 | 4.54 |
| Frequency | 1 | 2 | 10 | 39 | 49 | 4.32 |
| Travel time (vehicle speed) | 1 | 4 | 11 | 38 | 47 | 4.25 |
| Synchronization of transfers | 7 | 9 | 14 | 30 | 41 | 3.88 |
| A sense of security in means of transport and at stops | 1 | 6 | 16 | 32 | 46 | 4.15 |
| Availability of seats for journeys less than 20 min | 13 | 30 | 29 | 17 | 12 | 2.85 |
| Availability of seats for journeys longer than 20 min | 4 | 16 | 20 | 39 | 22 | 3.58 |
| Passenger information system in vehicles (voice announcements of stops, information screens) | 13 | 15 | 22 | 26 | 25 | 3.35 |
| Passenger information system at stops (information boards showing the time until the departure of vehicles on a given line) | 6 | 6 | 22 | 32 | 35 | 3.83 |
| Air conditioning/heating in the vehicle | 5 | 2 | 20 | 41 | 33 | 3.94 |
| Noise in the vehicle while traveling | 9 | 12 | 28 | 29 | 23 | 3.45 |
| Cleanliness of the vehicle and its interior | 3 | 3 | 24 | 34 | 37 | 3.98 |
| Access to additional amenities such as USB ports for charging your phone, access to WiFi | 24 | 26 | 26 | 15 | 10 | 2.61 |
| Age of vehicles, modern appearance, emission standard, ecological drive, etc. | 12 | 23 | 22 | 22 | 22 | 3.19 |
| Driver's driving style | 4 | 9 | 25 | 33 | 30 | 3.75 |
| Ticket system | 7 | 6 | 24 | 24 | 40 | 3.83 |
| Common ticket fare | 5 | 1 | 16 | 25 | 54 | 4.21 |
| Ticket prices | 6 | 0 | 9 | 16 | 70 | 4.43 |

Source: Elaboration based on K. Rusinek diploma thesis [2021].

As the table above shows, passengers attach the greatest importance to: certainty of reaching their destination, punctuality, ticket prices, frequency of courses, time (speed) of travel and a common ticket tariff. Passengers, in turn, pay the least attention to: additional amenities in the form of USB ports for charging phones or Wi-Fi, the availability of seats for journeys lasting less than 20 min, the age, appearance and ecological drive of vehicles and the passenger information system in vehicles.

The third part of the questionnaire concerned the same factors as the previous one, however, the respondents assessed the level of satisfaction with the current state of each of

the examined parameters of public transport. The respondents' assessments are presented in Table 2.

**Table 2.** Summary of the importance of the tested parameters for passengers and the calculated average weight.

| Public Transport Parameter/Feature | Respondents' Answers (Evaluation of Parameters) | | | | | |
|---|---|---|---|---|---|---|
| | 1 | 2 | 3 | 4 | 5 | Average Rating |
| Certainty of reaching the destination | 5 | 5 | 19 | 52 | 20 | 3.76 |
| Punctuality | 6 | 19 | 41 | 33 | 2 | 3.06 |
| Frequency | 6 | 23 | 49 | 21 | 2 | 2.90 |
| Travel time (vehicle speed) | 7 | 14 | 46 | 30 | 4 | 3.10 |
| Synchronization of transfers | 9 | 25 | 40 | 21 | 6 | 2.90 |
| A sense of security in means of transport and at stops | 7 | 13 | 39 | 36 | 6 | 3.21 |
| Availability of seats for journeys less than 20 min | 4 | 15 | 42 | 33 | 7 | 3..24 |
| Availability of seats for journeys longer than 20 min | 7 | 21 | 42 | 26 | 5 | 3.01 |
| Passenger information system in vehicles) | 7 | 18 | 31 | 33 | 12 | 3.25 |
| Passenger information system at stops | 5 | 13 | 23 | 41 | 19 | 3.55 |
| Air conditioning/heating in the vehicle | 10 | 20 | 44 | 22 | 5 | 2.92 |
| Noise in the vehicle while traveling | 7 | 23 | 49 | 19 | 3 | 2.88 |
| Cleanliness of the vehicle and its interior | 5 | 13 | 32 | 36 | 15 | 3.43 |
| Access to additional amenities such as USB ports for charging your phone, access to WiFi | 9 | 19 | 36 | 32 | 5 | 3.05 |
| Age of vehicles, modern appearance, emission standard, ecological drive, etc. | 5 | 15 | 38 | 38 | 5 | 3.23 |
| Driver's driving style | 6 | 10 | 31 | 41 | 13 | 3.45 |
| Ticket system | 13 | 12 | 26 | 34 | 16 | 3.28 |
| Common ticket fare | 9 | 10 | 27 | 31 | 24 | 3.50 |
| Ticket prices | 29 | 29 | 24 | 12 | 7 | 2.40 |

Source: Elaboration based on K. Rusinek diploma thesis [2021].

Analyzing the Table 2 ratings given by the members of the study to individual factors, it can be noticed that no parameter received an average score higher than 4. This is quite a disturbing phenomenon, considering that as many as 19 different features were analyzed. The highest ratings were given to: the certainty of reaching the destination, the passenger information system at stops, the common ticket tariff, the driving style of the drivers and the cleanliness of the vehicles and their interiors. The lowest scores were given by the respondents to: ticket prices, noise in vehicles, synchronization of transfers, frequency of journeys and air conditioning/heating in vehicles.

After collecting the answers and calculating the average values of weights and ratings for each of the examined factors (features and parameters of public transport), CSI indices were calculated. The results obtained for each of the factors as well as for the entire research object have been placed in Table 3.

The obtained results of the study show that the level of passenger satisfaction with the reliability of public transport is at an average level (ranges from 60 to 75%). This proves that there are reliability problems that should be solved or at least leveled.

Ticket prices are the most serious problem indicated by the respondents. The CSI index for this parameter is the lowest of all and amounts to only 48%, falling within the range indicating serious customer dissatisfaction. Other features assessed negatively are: noise in the vehicle, synchronization of transfers, frequency of trips, air conditioning/heating in the vehicle and availability of seats for trips lasting more than 20 min. Improving these parameters should be a priority for the research subject.

**Table 3.** Calculations and indicator values CSI oraz CSI %.

| Public Transport Parameter/Feature | Factor Evaluation $(c_i)$ | Factor Weight $(W_i)$ | Relative Weight $(W_{iw})$ | Indicator CSI $W_{iw} * c_i$ | CSI Max $W_{iw} * c_{imax}$ | CSI % |
|---|---|---|---|---|---|---|
| Certainty of reaching the destination | 3.76 | 4.64 | 0.064 | 0.240 | 0.319 | 75% |
| Punctuality | 3.06 | 4.54 | 0.062 | 0.191 | 0.312 | 61% |
| Frequency | 2.90 | 4.32 | 0.059 | 0.172 | 0.297 | 58% |
| Travel time (vehicle speed) | 3.10 | 4.25 | 0.058 | 0.181 | 0.292 | 62% |
| Synchronization of transfers | 2.90 | 3.88 | 0.053 | 0.155 | 0.267 | 58% |
| A sense of security in means of transport and at stops | 3.21 | 4.15 | 0.057 | 0.183 | 0.285 | 64% |
| Availability of seats for journeys less than 20 min | 3.24 | 2.85 | 0.039 | 0.127 | 0.196 | 65% |
| Availability of seats for journeys longer than 20 min | 3.01 | 3.58 | 0.049 | 0.148 | 0.246 | 60% |
| Passenger information system in vehicles | 3.25 | 3.35 | 0.046 | 0.150 | 0.230 | 65% |
| Passenger information system at stops | 3.55 | 3.83 | 0.053 | 0.187 | 0.263 | 71% |
| Air conditioning/heating in the vehicle | 2.92 | 3.94 | 0.054 | 0.158 | 0.271 | 58% |
| Noise in the vehicle while traveling | 2.88 | 3.45 | 0.047 | 0.137 | 0.237 | 58% |
| Cleanliness of the vehicle and its interior | 3.43 | 3.98 | 0.055 | 0.188 | 0.273 | 69% |
| Access to additional amenities such as USB Ports for charging your phone, access to WiFi | 3.05 | 2.61 | 0.036 | 0.109 | 0.179 | 61% |
| Age of vehicles, modern appearance, emission standard, ecological drive, etc. | 3.23 | 3.19 | 0.044 | 0.142 | 0.219 | 65% |
| Driver's driving style | 3.45 | 3.75 | 0.052 | 0.178 | 0.258 | 69% |
| Ticket system | 3.28 | 3.83 | 0.053 | 0.173 | 0.263 | 66% |
| Common ticket fare | 3.50 | 4.21 | 0.058 | 0.202 | 0.289 | 70% |
| Ticket prices | 2.40 | 4.43 | 0.061 | 0.146 | 0.304 | 48% |
| | | $\sum = 72.78$ | $\sum = 1$ | CSI **3.165** | CSI max **5** | CSI % **63%** |

Source: Elaboration based on K. Rusinek diploma thesis [2021].

The other features had a medium CSI (between 60 and 75%). What is important, the index of none of the parameters reached a good value, which proves that passengers perceive certain problems in each of the examined features.

The parameters rated the highest are: passenger information system at stops (71%), common ticket tariff (70%), driving style of drivers (69%) and cleanliness of vehicles and their interiors (69%). It is true that the values of the CSI indicators of these features indicate the fact that there is room for improvement (none of them reached 75%), however, taking into account the fact that there are more serious problems, the level of satisfaction with these parameters can be temporarily considered satisfactory, especially since the subject of the research has no significant impact neither on the passenger information system at stops nor on the ticket tariff, as these depend directly on the organizer of transport, i.e., the Metropolitan Transport Authority. Research subject as a public transport operator can only submit its comments on these issues, but it has no direct influence on them.

The last stage of the research using the CSI method was to create a quality map allowing to determine what strategy the company should adopt in relation to each of the examined parameters. The developed quality map is shown in Figure 4. The map has been scaled to ensure the best readability, but at the same time to be able to include all the examined factors.

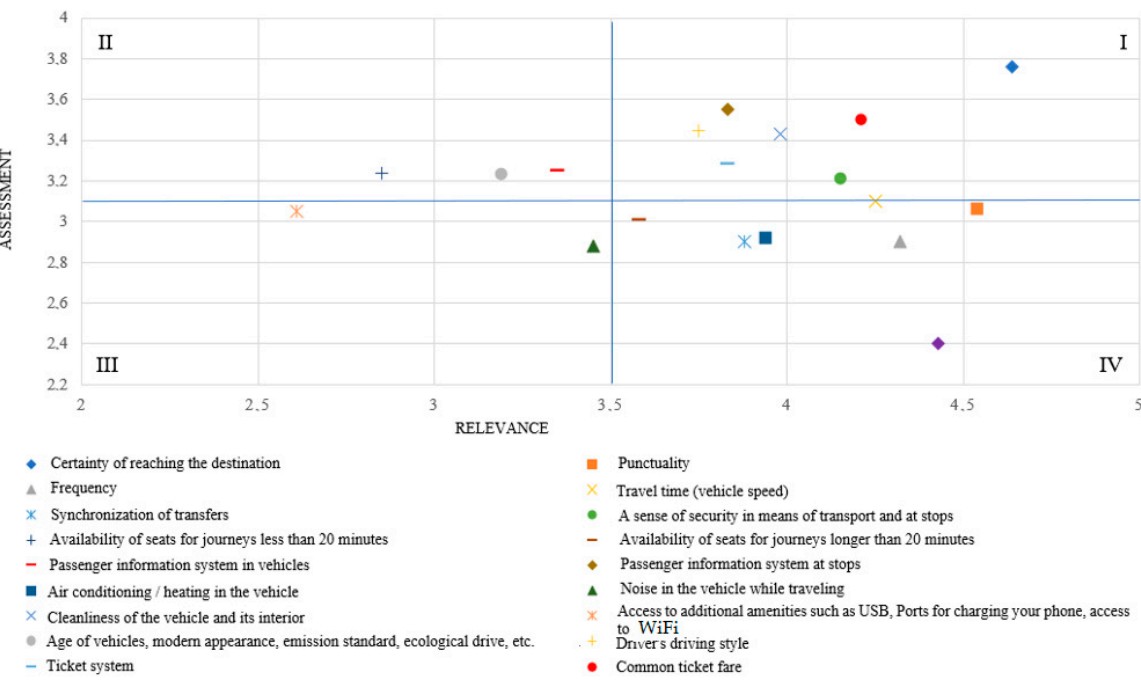

**Figure 4.** Quality map. Source: Elaboration based on K. Rusinek diploma thesis [2021].

The quality map has been divided into four areas, numbered according to the rules for numbering quadrants of the coordinate system. Individual parameters and features of public transport have been placed on the map according to their importance and assessment shared by the respondents of the study.

In the first quarter there are features that received a relatively high (compared to the rest of the factors) rating and are of great importance to passengers, these are:

- certainty of reaching the destination,
- passenger information system at stops,
- common ticket tariff,
- driving style of drivers,
- cleanliness of vehicles and their interiors,
- ticketing system,
- a sense of security in means of transport and at stops.

The enterprise should strive to maintain the current state of these parameters.

The second quadrant contains features that are rated relatively high but are of low importance to passengers. There are parameters such as:

- passenger information system in vehicles,
- age, modern appearance, emission standard and ecological vehicle drive,
- availability of seats for trips lasting less than 20 min.

These factors are not significant for passengers, and the level of satisfaction with their current level is satisfactory. This means that the company should not focus on them, but rather allocate resources to improve other features.

In the third quadrant are factors that received a relatively low rating, but they are also not very important for passengers. There are only two factors here:

- noise in the vehicle while traveling,
- additional amenities such as USB ports for charging your phone and access to WiFi.

The enterprise should strive to improve these parameters, but only after solving the more serious problems in quadrant four and making sure that their improvement will not cause deterioration of the characteristics in quadrant one.

The parameters that the company should focus on improving in the first place are in the fourth quarter. These are features of high importance to passengers, which at the same time received low ratings. This group of factors included:

- ticket prices,
- frequency of courses,
- air conditioning/heating in vehicles,
- synchronization of transfers,
- availability of seats for journeys lasting more than 20 min,
- punctuality,
- travel time/speed—this factor is located at the intersection between the 1st and 4th quadrants, therefore its improvement may be postponed in time if it would delay actions aimed at improving the remaining parameters in the 4th quadrant of the quality map.

Improving the above parameters should be a priority for the company. Unfortunately, some of them are beyond the control of the research subject. The best example of such a factor are ticket prices, which, although they are characterized by both the lowest CSI index and the worst position on the quality map, are not dependent on the decision of the research subject. The surveyed company as an operator can only report noticing a problem with this factor and possibly propose improvements, but the final opinion will be given to the Metropolitan Transport Authority, and in this matter (ticket prices) also the Metropolitan authorities, which determine, for example, social groups covered by discounts and the level of subsidizing public transport.

Other parameters in the fourth quarter on which the surveyed enterprise has a very limited influence are the synchronization of transfers and the frequency of trips. The timetables of lines are determined by the MTA and submitted for implementation to operators such as the surveyed company. Operators do not set timetables or organize courses on their own, they only service courses received from the MTA as part of a tender. However, they can submit their comments and proposals for changing the courses to the MTA in case of problems with their implementation.

A separate group of factors on which both the research subject and the transport organizer (MTA) have a very limited impact are the speed of travel and punctuality. The first of these factors most often results from road infrastructure not adapted to the conditions, unreasonable speed limits and changes aimed at forcing the priority of pedestrian and bicycle traffic at the expense of slowing down the traffic of motor vehicles. Such measures affect not only passenger cars, but also public transport, extending the travel time and thus causing a decrease in passenger satisfaction. The enterprise may submit its comments regarding the infrastructure to the relevant authorities, but it has no influence on their consideration and implementation. The lack of punctuality is also related to the limited capacity of the infrastructure, as the most common cause of delays is transport congestion. This may result from min. fortuitous events (collision, accident), road works, but most often it is caused by too much traffic due to insufficient road infrastructure capacity, which is easy to observe during peak hours. However, the surveyed company has no influence on the policy of the city authorities forming the Metropolis (striving to combat and slow down the traffic of motor vehicles in cities by force), or on the condition and capacity of the infrastructure (insufficient with heavy traffic). This does not mean, however, that there are no actions that could improve these parameters, but they would have to be introduced in consultation with the transport organizer, as they would require changing the routes of some bus lines so that they avoid areas particularly exposed to congestion and characterized by artificially limited travel speeds. However, such actions must be undertaken carefully, as they may adversely affect the accessibility of some areas by transport. A reasonable compromise would be to introduce more fast and fast lines that would avoid the most problematic areas, while maintaining the functioning of regular lines, whose role would focus mainly on transporting people to stops served by fast and fast lines. For such a

solution to work, however, it would be necessary to ensure a high frequency of courses, which would entail high costs for the organizer.

The group of factors that should be improved, however, also included parameters on which the research subject has a direct and total impact, such as the availability of seats for journeys lasting more than 20 min and air conditioning/heating in vehicles. These factors can be improved by the company on its own, because they are fully dependent on it, and therefore it is on them that the surveyed company should focus and, if possible, allocate funds to improve them.

## 5. Conclusions

Public transportation issues are one of the most common problems in all countries. Most studies focus mainly on travel time, which leads to neglecting other aspects.

The term transportation reliability was created to cover, study and analyze all transportation issues.

The research conducted using the CSI method allowed to find out the opinion of passengers on public transportation in the GZM in terms of selected characteristics (transportation demands).

Residents, by means of a questionnaire, gave weight to individual transportation postulates and assessed the current degree of their implementation. The value of the CSI index for public transportation in the GZM indicates an average level of passenger satisfaction with its reliability, which means that the company should take measures to improve it. The value of the CSI index for public transportation in the Metropolis indicates an average level of passenger satisfaction with its reliability, which means that the enterprise should take measures to improve this condition. Identification of the parameters on which the re-study entity should focus first was made possible by the quality map prepared.

However, it should be noted that many of them (such as fare prices) are beyond the company's control, as they depend on the decisions of the MTA, the Metropolitan Authority or the authorities of the constituent cities. The company under study, as a public transport operator acting on behalf of the MTA, can only propose improvements in this area, but is not responsible for implementing them. Among the parameters to be improved, however, there were also those that the surveyed company has full influence over and that are dependent on its operations. It is their improvement that the company should focus on first, allocating resources, where possible, to activities aimed at improving them.

This article is designed to shed light on the reliability of public transportation and show the need for further research in this aspect to provide, according to M.M. Cahigas et al. [44] practical implications for public transportation managers to adapt to the needs of passengers.

**Funding:** Publication supported under the Excellence Initiative—Research University program at the Silesian University of Technology, year 2022, 13/050/SDU/10-22-01.

**Institutional Review Board Statement:** Not applicable.

**Informed Consent Statement:** Not applicable.

**Data Availability Statement:** Not applicable.

**Conflicts of Interest:** The author declares no conflict of interest.

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
