# Peer review of "The Problem of Reliability in Public Transport for the Metropolis GMZ Area-Pilots Studies"

_sustainability, doi:10.3390/su15043199_

Round 1

Reviewer 1 Report

Thank you the opportunity to review the sound paper presented. The results and background of the study gave me much interest in reading the paper. However upon further reader, I believe that the study needs further improvement for emphasis and reach wider audiences.

1. I would suggest authors to argue more first the analysis of focusing solely on reliability when most transportation studies have utilized the SERVQUAL dimensions or expanded/modified service quality framework. Ergo discuss studies relating to these and then justify reliability as sole analysis. For example, studies such as that of Chuenyindee et al. (2022) focused on SERVQUAL dimensions and justified new environment should be reanalyzed and so on (https://www.sciencedirect.com/science/article/pii/S0957178722000029).

2. The title and abstract should indicate the pilot study's location as the current state projected a generalized findings, but is a case study of the output.

3. Discussions were made based on results, but no arguements of contrats with other study findings were indicated.

4. With that comparison of which should also be made.

5. Please provide your method questions as how you disseminated the questionnaires in the appendix section. I would suggest that supporting references would be placed in your current methodology such as adaption of questionnaires and such. 

6. Practical implication, theoretical implications, and government implications or managerial insights I believe would provide more solutions in the study which could be a basis for other studies and implementation of the findings. This contribution would be highly relevant in application. Reference from Cahigas et al. (2022) with the contribution suggestions (https://www.sciencedirect.com/science/article/pii/S221053952200061X).

7. Moreso, the sustainability aspect may be considered by relating your study to sustainable transportation,management, and cities such as that of German et al. (2022). (https://www.sciencedirect.com/science/article/pii/S2405844022026706).

8. Please apply proper MDPI-Sustainability formatting from structure to references.

I hope that the suggestions made would be appreciated and highlight the contribution of the study. I hope to cite your paper soon.

Author Response

Thank you for your very valuable comments in your review. 
Enclosed is the revised article as suggested in the review. 
In addition, the article has been corrected according to the comments of the editor and other reviewers

Reviewer 2 Report

My comments for the paper with the title “The problem of reliability in public transport - pilots studys” are below:

·        Every introduction should present the issue that the study is addressing the approach that the study is following for addressing the issue, the novelty of the study, some of the most important results, and a short conclusion based on the outcomes. This introduction is very short for a journal.

·        On lines 43, 90, and 119 there is bold font. These are sub-sections? This is not the correct format for sub-sections.

·        In the theoretical sections, the author should present the contribution of this study to the existing literature.

·        The references in this paper are very few. The literature review must be extended.

·        The author should present, with details, the methodology used in this paper and not the percentage of respondents, which should be presented in the results. In this, section it should be clear to the reader which method was used, and the reason for using this method.

·        The author should present descriptive statistics of the survey in the Appendix.

·        What is the sample size of the survey? Where does it take place? Is it considered one area or does it cover the entire public transport network?

·        The author should review the paper for typos and syntax errors. For example, section 4 should be “Results” and not “Resaults”. Even, the title of the paper is wrong: “studies” and not “studys”.

·        The figures should be improved considering their resolution

·        Conclusions should present the most significant outcomes of the analysis. In Figure 4, what is the x and y axis representing? They are not in English.

Author Response

Thank you for your very valuable comments in your review. 
Enclosed is the revised article as suggested in the review. 
In addition, the article has been corrected according to the comments of the editor and other reviewers.

Round 2

Reviewer 1 Report

The author was able to apply the constructive comments and suggestion. Thank you very much.

Reviewer 2 Report

The author has addressed all my comments and thus I recommend this paper for publication